# Comprehensive Comparative Analyses of *Aspidistra* Chloroplast Genomes: Insights into Interspecific Plastid Diversity and Phylogeny

**DOI:** 10.3390/genes14101894

**Published:** 2023-09-29

**Authors:** Jie Huang, Zhaocen Lu, Chunrui Lin, Weibin Xu, Yan Liu

**Affiliations:** 1Guangxi Key Laboratory of Plant Functional Phytochemicals and Sustainable Utilization, Guangxi Institute of Botany, Guangxi Zhuang Autonomous Region and Chinese Academy of Sciences, Guilin 541006, China; huangjie0773@foxmail.com (J.H.); zhaocenlu@163.com (Z.L.); gxibly@163.com (Y.L.); 2Guangxi Key Laboratory of Plant Conservation and Restoration Ecology in Karst Terrain, Guangxi Institute of Botany, Guangxi Zhuang Autonomous Region and Chinese Academy of Sciences, Guilin 541006, China

**Keywords:** *Aspidistra*, chloroplast genome, interspecific diversity, plastome evolution, phylogenomics

## Abstract

Limestone karsts are renowned for extremely high species richness and endemism. *Aspidistra* (Asparagaceae) is among the highly diversified genera distributed in karst areas, making it an ideal group for studying the evolutionary mechanisms of karst plants. The taxonomy and identification of *Aspidistra* species are mainly based on their specialized and diverse floral structures. *Aspidistra* plants have inconspicuous flowers, and the similarity in vegetative morphology often leads to difficulties in species discrimination. Chloroplast genomes possess variable genetic information and offer the potential for interspecies identification. However, as yet there is little information about the interspecific diversity and evolution of the plastid genomes of *Aspidistra*. In this study, we reported chloroplast (cp) genomes of seven *Aspidistra* species (*A. crassifila*, *A. dolichanthera*, *A. erecta*, *A. longgangensis*, *A. minutiflora*, *A. nankunshanensis*, and *A. retusa*). These seven highly-conserved plastid genomes all have a typical quartile structure and include a total of 113 unique genes, comprising 79 protein-coding genes, 4 rRNA genes and 30 tRNA genes. Additionally, we conducted a comprehensive comparative analysis of *Aspidistra* cp genomes. We identified eight divergent hotspot regions (*trnC*-GCA-*petN*, *trnE*-UUC-*psbD*, *accD-psaI*, *petA-psbJ*, *rpl20-rps12*, *rpl36-rps8*, *ccsA-ndhD* and *rps15-ycf1*) that serve as potential molecular markers. Our newly generated *Aspidistra* plastomes enrich the resources of plastid genomes of karst plants, and an investigation into the plastome diversity offers novel perspectives on the taxonomy, phylogeny and evolution of *Aspidistra* species.

## 1. Introduction

Limestone karsts, with their unique physical and chemical substrates, shape a range of microhabitats and unique island habitats, such as karst caves, and terrestrial islands [1]. These special habitats have ecological conditions that significantly differ from the surroundings and thus provide relatively isolated environments for species isolation and differentiation [2,3]. Karst areas have harboured a high level of plant richness and endemism through a long period of evolution [2]. For example, the limestone ecosystem has reported plant groups with rich and marvellous flora, such as Gesneriaceae and *Begonia*, that exhibit prominent morphological variation [4,5,6]. Many karst plant species often have very narrow distribution ranges. However, damage to the fragile ecological environments threatens the karst plants partly because of growing human activities (tourism industry, etc.). Thus, accurate species identification is particularly important for the utilization and protection of these plants.

*Aspidistra* (Asparagaceae) is one of the represented plant groups of evergreen perennials in limestone karsts, with a center of diversity in southwest China and northern Vietnam [7]. Over the last forty years, the number of recognized *Aspidistra* species has experienced a remarkable increase [8], rapidly rising from 11 (until the year 1978) [9] to more than 200 [10,11,12,13]. However, the diversity of *Aspidistra* is still far from being revealed, with more new species being discovered during field investigations every year [8,11,12,14]. Despite its richness in species, *Aspidistra* plants are morphologically similar in vegetative characters (such as habit and leaf, etc.). Most *Aspidistra* species are characterized by creeping rhizomes, except several members which have an erect stem (e.g., *A. erecta* Yan Liu & C.I Peng and *A. brachypetala* C.R. Lin & B. Pan) [15]. The species of *Aspidistra* can be classified into two main groups based on their leaf growth patterns. One group has solitary and elliptical (oblong) or oval-shaped leaves with petioles (e.g., *A. elatior* Bl., *A. austroyunnanensis* G.W. Hu, Lei Cai & Q.F. Wang and *A. chunxiusis* C.R. Lin & Yan Liu) [16,17,18]. In contrast, the other group possesses linear leaves lacking a petiole or indistinctly differentiated from leaf blade (e.g., *A. longifolia* Hook.f. and *A. hainanensis* W.Y. Chun & F.C. How) [10,19]. In most cases, accurate identification of *Aspidistra* is almost impossible in non-flowering conditions [10]. Interspecific identification of the genus relies mainly on remarkably varied floral structures. However, most *Aspidistra* species have cryptic flowering and fruiting. The inconspicuous *Aspidistra* flowers, commonly found near the ground and concealed beneath leaf litter [20,21], can be easily overlooked or difficult to collect during field investigations, which might have hindered identifying species within the genus [22]. In addition, even supplemented with evidence of cytophylogenies (e.g., chromosome number, size, and morphology of chromosome), micromorphological features and palynological characters (pollen), these are far from sufficient for species identification of the genus [10,23].

DNA barcoding is considered a reliable and cost-efficient method for authenticating plants. Molecular markers in plants hitherto mainly come from chloroplast DNA (cpDNA). The cpDNA is a DNA fragment derived from the chloroplast (cp) genome, which encodes proteins involved in photosynthesis and generally has unique characteristics. The cp genome is usually maternal inherited and exhibits a high level of conservation in its genome structure and coding sequences. With a size range of 120–160 kb, the cp genome is composed of four distinct regions, namely a pair of inverted repeat sequences (IRs), a large single-copy (LSC) and a small single-copy (SSC) region. The plastid genome offers valuable information for examining evolution, DNA barcoding, taxonomy, and phylogeny [24]. However, some frequently used plastid regions (such as *psbA-trnH*, *matK*, *psbK-psbI* and *rbcL*, etc.) are insufficient to discriminate *Aspidistra* at the species level [25]. Therefore, few DNA molecular markers have been suitable for identifying *Aspidistra* species until now.

With the development of high-throughput sequencing technologies, plastid genomes (plastomes) and genome-wide nuclear sequence data of numerous plant species have been sequenced and applied in phylogenetic studies. As for the Asparagaceae, cp genomes of over 200 species have been reported (data from CGIR (Chloroplast Genome Information Resource)) [26,27,28,29]. Although four plastomes of *Aspidistra* species are available (accessed on 1 August 2023) in the National Center for Biotechnology Information (NCBI), there is still a lack of information regarding the plastid phylogenomics of the genus. To address this gap, we conducted a study in which we newly sequenced seven *Aspidistra* chloroplast genome sequences. Additionally, we conducted a comparative analysis of these cp genomes with the previously available four *Aspidistra* plastomes. We investigated the general plastome features, structural characteristics and sequence divergence of the *Aspidistra* plastomes. We identified highly variable regions within the cp genomes that could potentially serve as barcodes for species identification. Moreover, we utilized the complete plastome sequence of the genus to infer the phylogenetic relationships. Our comprehensive examination of chloroplast genomes provides a valuable theoretical foundation for future taxonomic and evolutionary studies of *Aspidistra* plants and helps fill the knowledge gap regarding the plastid phylogenomics of the genus.

## 2. Results

### 2.1. Characteristics of Chloroplast Genome Sequences

We sequenced and assembled seven chloroplast genomes of *Aspidistra* species (Appendix A). All the *Aspidistra* complete cp sequences were typical circular DNA molecules, ranging from 156,335 to 156,475 bp (Figure 1 and Table 1). They were conserved in length and had a typical plastid genome structure, consisting of a pair of IR regions (26,498 to 26,540 bp), and one LSC (85,102 to 85,181 bp) and SSC region (18,199 to 18,232 bp). The average GC contents of the genomes ranged from 37.61% to 37.65%, with a significantly higher value in the IR regions (42.95–43%) than in the LSC (35.54–35.59%) and SSC (31.61–31.74%) regions. The chloroplast genomes are available in the NCBI database with accession numbers listed in Table 1.

### 2.2. Genome Annotation

The genomic structures of the seven cp genomes exhibited a high level of conservation, with the same gene numbers, composition, order and orientation. Table 1 presents the fundamental genomic characteristics of the *Aspidistra* species. The seven plastid genomes all contain 113 unique genes, including 79 protein-coding genes, 4 rRNA genes and 30 tRNA genes (Appendix A).

Seven protein-coding genes (*rpl2*, *rpl23*, *rps12*, *rps19*, *rps7*, *ndhB* and *ycf2*), four rRNA (*rrn16S*, *rrn23S*, *rrn5S* and *rrn4.5S*) and eight tRNA genes (*trnA*-UGC, *trnH*-GUG, *trnI*-CAU, *trnI*-GAU, *trnL*-CAA, *trnN*-GUU, *trnR*-ACG and *trnV*-GAC) were located in the IR regions, resulting in two copies of these genes. The chloroplast genome contained a total of 17 genes that had introns. Out of the identified genes, there were 11 protein-coding genes (*rps16*, *atpF*, *rpoC1*, *ycf3*, *clpP*, *petB*, *petD*, *rpl16*, *rpl2*, *ndhB*, and *ndhA*) and six tRNA genes (*trnK*-UUU, *trnG*-UCC, *trnL*-UAA, *trnV*-UAC, *trnI*-GAU and *trnA*-UGC). The trans-spliced gene *rps12* has two copies, with the 5′ end located in the LSC region and the 3′ ends in the IRa and IRb regions, respectively.

### 2.3. Repeat Sequences

The total numbers of simple sequence repeats (SSRs) we identified for the sampled *Aspidistra* species were as follows: 62 (*A. minutiflora*), 57 (*A. retusa*), 60 (*A. dolichanthera*), 64 (*A. crassifila*), 66 (*A. erecta*), 66 (*A. longgangensis*), and 67 (*A. nankunshanensis*) (Figure 2A and Appendix A). Most SSRs in the seven *Aspidistra* plastid genomes were mononucleotide repeats, with numbers of repeats ranging from 29 (*A. retusa*) to 37 (*A. longgangensis* and *A. nankunshanensis*), accounting for 50.88–56.06% of the total SSRs (Figure 2A). Other types of SSRs included dinucleotide repeats (range in number of repeats and proportion of all SSRs of these *Aspidistra* plastomes: 12–14, 18.46–21.67%), trinucleotide repeats (5, 7.46–8.77%), tetranucleotide repeats (9–10, 13.64–15.79%), and pentanucleotide repeats (2–3, 2.99–4.84%) (Figure 2A and Appendix A). No hexanucleotide repeats were found in any *Aspidistra* plastomes.

In addition to SSRs, we also detected four other interspersed repeats (forward, reverse, palindromic, and complement repeats) (Figure 2B). Among them (37–47 repeat sequences for seven chloroplast genomes), most were forward repeats (17–21; 44.19–47.37%) and palindromic repeats (19–24; 47.73–52.63%). There were two reverse repeats in *A. erecta*, while only one was found in the other six *Aspidistra* species. Complement repeats were found in four *Aspidistra* species (two repeats in *A. longgangensis*; one in *A. retusa, A. crassifila* and *A. nankunshanensis*).

### 2.4. Genomic Divergence

Genomic variation in the seven *Aspidistra* species was examined in mVISTA [30,31], with the cp genome of *Aspidistra minutiflora* as a reference sequence (Figure 3). The plastomes were highly conserved, but some different levels of sequence variation were still observed, particularly in the LSC and SSC regions (Figure 3 and Figure 4). Most high-variable sites belonged to the conserved noncoding sequences (CNS) regions.

We detected DNA polymorphisms using DnaSP v6.0 [32] based on the 11 aligned *Aspidistra* chloroplast genomes (including the other four *Aspidistra* plastomes available in NCBI) (Figure 4). The *Aspidistra* cp genomes were highly conserved. The nucleotide diversity (Pi) values of *Aspidistra* species (500 bp sliding widows) ranged from 0 to 0.013 (average value of 0.002394), with relatively high variation rates in the LSC and SSC regions and low rates in the two IR regions. There were eight positions with Pi values ≥ 0.007. Among them, six regions were located in the LSC region (*trnC*-GCA-*petN* (0.01033), *trnE*-UUC-*psbD* (0.01171), *accD-psaI* (0.00796), *petA-psbJ* (0.00898), *rpl20-rps12* (including two adjacent sliding windows with Pi values of 0.00771 and 0.00764, respectively) and *rpl36-rps8* (0.00727)), and the other two were in the SSC region (*ccsA-ndhD* (0.01207) and *rps15-ycf1* (0.00749)).

### 2.5. Contraction and Expansion of IR Regions

Plastid genomes of the *Aspidistra* species were highly conserved, showing very limited variations of SC/IR boundaries (Figure 5). Several genes (such as genes *rpl22*, *rps19*, *ndhF*, *ycf1*, *rpl2*, *trnH* and *psbA*) were found spanning across or near the junctions of the IR and SC regions. *rps19* and *trnH* both had two copies and completely accessed the two different IR (IRa and IRb) regions. The *rps19* gene was located at the boundary between the LSC and IR regions, with a distance of 45 to 47 bp from the border. Similarly, the *rpl22* was found within 8 bp to 10 bp of the JLB line. In all 11 *Aspidistra* plastomes, *ndhF* crossed the IRb/SSC boundary, with 29 bp in the IRb region and the remaining 2182 bp in the SSC region. The *ycf1* gene was located in the boundary between the SSC and IRa regions, with 4486–4510 bp of its length in the SSC region and 899 bp in the IRa region. In all 11 chloroplast genomes, a ycf1 pseudogene is present in the IRa region, with a corresponding fragment ranging from 898 bp to 901 bp in the IRb region.

### 2.6. Phylogenetic Relationships

Phylogenetic analyses were performed in IQ-TREE version 2.2.0.3 using maximum likelihood (ML) methods. We sampled a total of 40 representative species from three subfamilies of Asparagaceae, namely, Nolinoideae (36 species), Asparagoideae (two species), and Lomandroideae (two species). The phylogenomic analysis (Figure 6) recovered the six tribes in subfamily Nolinoideae as major clades with strong bootstrap (BS): Convallarieae (19 species, BS = 100%), Ophiopogoneae (five species, BS = 100%), Nolineae (two species, BS = 100%), Polygonateae (seven species, BS = 100%), Draceaneae (two species, BS = 100%) and Ruscineae (with only one representative species). The monophyly of the genus *Aspidistra* was strongly supported (BS = 100%), with *A. erecta* as the earliest branching species. Four species were successively diverging, namely, *A. nankunshanensis*, *A. retusa*, *A. elatior* and *A. crassifila*. The remaining six species form a clade (BS = 81%) containing three subclades (all subclades BS = 100%), with two species of each subclade, namely, *A. yingjiangensis* and *A. minutiflora*, *A. cavicola* and *A. dolichanthera*, and *A. obliquipeltata* and *A. longgangensis*.

## 3. Discussion

### 3.1. Chloroplast Genome Evolution with Conserved Genome Structure

We reported and analyzed the seven cp genomes of the genus *Aspidistra*, which have lengths ranging from 156,335 to 156,475 bp and a typical tetrad structure. These *Aspidistra* plastomes showed similarities to those of other members in the Asparagaceae family. We observed no gene gain and loss in the seven plastomes, suggesting that cp genomes were conserved in *Aspidistra* plants. Previous research has highlighted unique evolutionary events in the chloroplast genomes of certain Asparagaceae species, such as the incorporation of mitochondrial DNA sequences into the plastid genome in the *Convallaria* genus [33,34]. Expansion and contraction of the IR regions are commonly observed in angiosperms, as documented in previous studies [35,36]. In some cases, certain species have even experienced the complete loss of an IR region [37,38]. Our comparative analysis revealed that the IR regions exhibited no significant variation among the seven *Aspidistra* cp genomes, with lengths ranging from 26,498 bp to 25,540 bp.

### 3.2. Repeat Sequence Analysis

In the study, we discovered a range of 57 (*A. retusa*) to 67 (*A. nankunshanensis*) SSRs in seven different species of *Aspidistra*. Most of these SSRs were mononucleotide repeats, making up 50.88% to 56.06% of all the SSRs, similar to that in other angiosperm chloroplast genomes [39,40]. Most mononucleotide repeats were A/T repeats, generally polyadenine (Poly-A) or polythymine (Poly-T) repeats, and thus significantly impacted the overall G/C content of chloroplast genomes [41]. The lengths of SSRs were shown to be polymorphic in different species and are thought to be potential markers in species identification. Huang et al. [42] developed 48 SSR loci based on the transcriptome data from the flowers of *Aspidistra saxicola*. They discovered that approximately a third of the loci were polymorphic in four distinct populations of *A. elatior.* Our research also identified a significant number of SSRs within the *Aspidistra* cp genomes (Appendix A), which might also be valuable sources for developing markers to identify species within the genus.

### 3.3. Phylogenetic Relationship Based on the Plastid Genome

The complete chloroplast genome has been frequently employed in resolving the phylogenetic relationships of plants across various classification levels [3,29,43,44,45,46]. This study used 40 Asparagaceae species (including 36 species for the subfamily Nolinoideae; see Section 4.4) to construct the ML tree based on plastid genomes. Our results provide a well-resolved phylogenetic framework for the subfamily Nolinoideae (Figure 6). The first diverging clade (BS = 100%) consists of two tribes (Draceaneae and Ruscineae). The remaining Nolinoideae species are divided into two clades: one (BS = 100%) containing the tribe Convallarieae; and the other (BS = 100%) encompassing the tribes Ophiopogoneae, Nolineae, and Polygonateae. The phylogenetic relationships among these six tribes were consistent with the Nolinoideae phylogeny from plastid phylogenomics [29]. The monophyly of *Aspidistra* was well supported with 100% BS (Figure 6). In our results, the sister genus of *Aspidistra* was *Tupistra*, a finding similar to those of previous studies based on the plastid genome (containing four *Aspidistra* species) [29] and transcriptome data (with one *Aspidistra* species) [47]. Within the *Aspidistra*, the first branching clade was *A. erecta*, representing a group with distinguishable vegetative morphology (erect stem) [10,48]. After the divergence of *A. nankunshanensis* and *A. retusa*, the branching order of *A. elatior*, *A. crassifila* and a clade with the remaining six *Aspidistra* species was not well defined (BS = 48% and 43%); the six *Aspidistra* species group into three well-supported subclades (each containing two species), and the result can also be supported by evidence from morphology. The subclade of *A. yingjiangensis* and *A. minutiflora* represents a group of *Aspidistra* species with tufted (clustered) linear leaves [49,50]. *A. cavicola* and *A. dolichanthera* have a white, wide campanulate (broad bell-shaped) to rotate perianth [51,52]; while the perianth tube was urceolate in the subclade including *A. obliquipeltata* and *A. longgangensis* [53]. In conclusion, the chloroplast genome sequences offered valuable genetic information regarding the phylogenetic relationships of *Aspidistra*. However, the results are somewhat limited due to the plastid genome’s highly conserved nature and maternal inheritance property. Furthermore, more species needed to be sequenced to determine the relationships among *Aspidistra*.

### 3.4. Eight Highly Variable Cp Genome Regions Show Great Potential as DNA Barcodes

The chloroplast genomes of plants exhibit a high degree of conservation, making DNA fragments within these genomes effective tools in phylogenetic and evolutionary analyses. The complete chloroplast sequence has recently been utilized as a super-barcode to accurately identify and distinguish plant species [54,55].

As for the *Aspidistra*, one previous study [25] tested the efficiency of four universal DNA barcodes (*matK*, *psbA-trnH*, *psbK-psbI* and *rbcL*) in 19 species of the genus, and the individual successful identification rates of the barcode sequences ranged from 43.6% (*psbA*-*trnH*) to 88.7% (*rbcL*). Using a combination of markers had a higher identification efficiency than that of single sequences. The identification success rate using *matK* + (*psbK*-*psbI*) reached 100% in the 19 *Aspidistra* species, while the identification rates of these two sequences individually were 78% (*matK*) and 88.7% (*psbK*-*psbI*) [25]. Using this sequence combination (i.e., *matK* + (*psbK*-*psbI*)) also resulted in a 100% success identification rate in another study with 11 *Aspidistra* species [56].

With the growing number of identified *Aspidistra* species (currently exceeding 200), it became necessary to test the discriminatory power of these DNA markers in a broader range of species within the genus. In addition, new and more efficient DNA barcodes are necessary for the *Aspidistra* species. We evaluated the nucleotide diversity of the cp sequence using a sliding window analysis and detected eight DNA loci having Pi values greater than 0.007. The hypervariable regions were *trnC*-GCA-*petN* (Pi = 0.01033), *trnE*-UUC-*psbD* (Pi = 0.01171), *accD-psaI* (Pi = 0.00796), *petA-psbJ* (Pi = 0.00898), *rpl20-rps12* (Pi = 0.00771 and 0.00764 for the two adjacent sliding windows, respectively), *rpl36-rps8* (Pi = 0.00727), *ccsA-ndhD* (Pi = 0.01207) and *rps15-ycf1* (Pi = 0.00749)*,* indicating their potential to be developed as molecular markers. The previously used four barcodes (*matK*, *psbA-trnH*, *psbK-psbI* and *rbcL*) were not among the eight highly variable regions we detected, suggesting a possible higher identification efficiency of the potential markers. These findings are significant for identifying, protecting and rationally utilizing *Aspidistra* plants in karst areas.

## 4. Materials and Methods

### 4.1. Plant Sampling and Data Collection

We procured fresh leaves from seven *Aspidistra* species cultivated at the Guangxi Institute of Botany, Chinese Academy of Sciences, Guilin, China. Professor Chunrui Lin (Guangxi Institute of Botany) conducted the formal identifications of all samples. The selected *Aspidistra* species included members with different types of forms or morphologies and geological distribution regions. For example, we sampled species with erect stems (*A. erecta*) or creeping rhizomes (*A. nankunshanensis*, *A. retusa*, etc.), as well as species with clustered linear leaves (*A. minutiflora*) or solitary oblong leaves with petioles (*A. obliquipeltata* and *A. longgangensis*). The sampled taxa also included members with high ornamental value (having unique flowers), such as *A. dolichanthera*, *A. longgangensis* and *A. crassifila*. All sampled vouchers were deposited in the Herbarium, Guangxi Institute of Botany (IBK; Guilin, China), with detailed information and deposition numbers shown in Supplementary Infomation (Appendix A). The total genomic DNA of each sample was extracted following a modified CTAB method [57]. We then constructed a DNA library for each sample with 350 bp inserted fragments following the Illumina NovaSeq protocol. Raw reads were sequenced in the Illumina Nova6000 platform. After eliminating low-quality reads, clean data were obtained. The rest of the 33 plastomes (including four *Aspidistra* species and 29 representatives in the family Asparagaceae) were obtained from the NCBI.

### 4.2. Genome Annotation

Clean data were de novo assembled into the cp genome using GetOrganelle [58]. The average coverage depth of *Aspidistra* plastomes ranged from 63 to 512× (Appendix A). Initially, the gene annotation of each plastome was implemented in CpGAVAS2 [59] with *Reineckea carnea* as a reference (GenBank: MK801116), then manually corrected in Apollo genome editor v1.11.8 [60]. Necessary corrections included the identification of missing genes, checking and adjusting the start and stop positions of genes, annotating small exons for some genes (such as *petB*, *petD*, *rpl16*, etc.), and curating gene structures (such as *rps12*). We utilized the OGDRAW program to create genome maps [61]. The seven *Aspidistra* chloroplast genomes have been submitted and are now available in the GenBank database (Table 1).

### 4.3. Repeats Sequences Analysis and Genome Comparison

The GC content of the plastid genomes was estimated using EMBOSS [62]. SSRs of each plastid genome were detected on the website MISA “https://webblast.ipk-gatersleben.de/misa/ (accessed on 9 June 2023)”, including mononucleotides (with a minimum number of repetitions ≥10), dinucleotides (≥5), trinucleotides (≥4), tetranucleotides (≥3), pentanucleotides (≥3) and hexanucleotides (≥3). Other repetitive sequences (i.e., palindromic repeats, forward repeats, reverse repeats, and complementary repeats) were identified by REPuter “https://bibiserv.cebitec.uni-bielefeld.de/reputer/ (accessed on 10 July 2023)” with parameters being 3 (hamming distance) and of 30 bp (minimum repeat size).

The seven *Aspidistra* (*A. minutiflora*, *A. retusa*, *A. dolichanthera*, *A. crassifila*, *A. erecta*, *A. longgangensis*, and *A. nankunshanensis*) plastomes were compared in mVISTA [30,31] to explore interspecific variations and identify hotspot regions “http://genome.lbl.gov/vista/mvista/submit.shtml (accessed on 7 August 2023). The junction boundaries of these chloroplast genomes were visualized in IRscope [63] “https://irscope.shinyapps.io/irapp/ (accessed on 10 August 2023). Four other publicly available *Aspidistra* plastomes (*A. yingjiangensis*, *A. cavicola*, *A. obliquipeltata* and *A. elatior*) and three outgroup plants (*Tupistra grandistigma*, *Rohdea chinensis* and *Reineckea carnea*) were also included in the analysis to compare junction boundaries. The nucleotide polymorphism (Pi) among the 11 *Aspidistra* species was calculated by DnaSP [32] using the aligned chloroplast sequences matrix, with the sliding window length being 500 bp.

### 4.4. Phylogenetic Analysis

In addition to 11 *Aspidistra* species, including four other publicly available *Aspidistra* plastomes (*A. yingjiangensis*, *A. cavicola*, *A. obliquipeltata* and *A. elatior*), a total of 40 Asparagaceae species were included in the phylogenetic analyses. These species belong to three subfamilies, i.e., Lomandroideae (two species), Asparagoideae (two species), and Nolinoideae (36 species), with Nolinoideae species from tribes of Convallarieae (19 species), Ophiopogoneae (five species), Polygonateae (seven species), Nolineae (two species), Draceaneae (two species) and Ruscineae (one species). The 40 complete plastome sequences were aligned in MAFFT [64] with default parameters and then used to infer the phylogenetic tree implemented in IQ-TREE multicore version 2.2.0.3 [65] with the ML method (1000 bootstrap replicates).

## 5. Conclusions

In this study, we reported the chloroplast genomes of seven *Aspidistra* species and conducted comprehensive analyses of these plastid genomes with the other available plastomes within the genus. Each *Aspidistra* plastome has a typical quartile structure similar to that in most angiosperms, and is highly conserved in regard to GC content, gene number, order and structural characteristics. A comprehensive analysis of these plastomes shows limited interspecific diversity within the genus. In addition, we identified eight relatively hypervariable regions of the *Aspidistra* chloroplast genome as candidate markers. These analyses of chloroplast genomes present a new perspective into the taxonomy, identification, phylogeny and evolution of *Aspidistra* species, providing a basis for the utilization and conservation strategy of this representative karst plant group.

## Figures and Tables

**Figure 1 genes-14-01894-f001:**
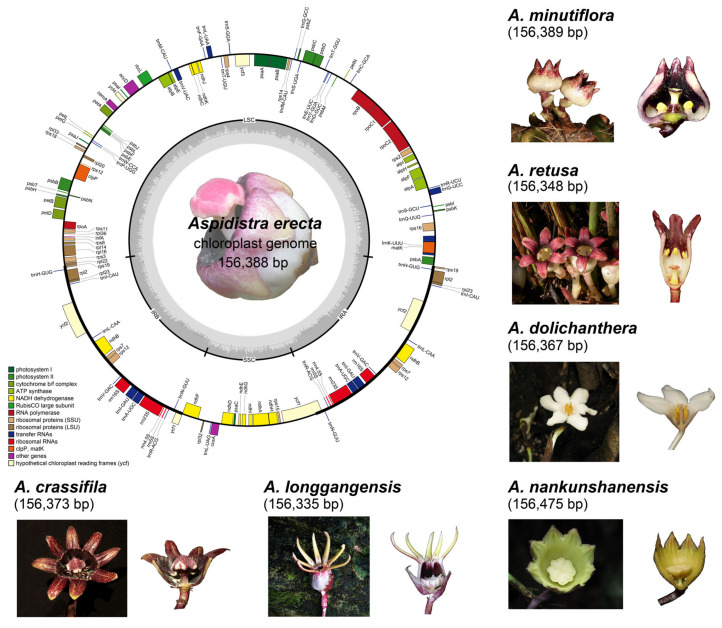
Structural map and images of seven *Aspidistra* species. All seven plastid genomes possess the characteristic quartile structure, comprising a pair of IR regions, an LSC region, and an SSC region. The genes inside the circle are transcribed in a clockwise direction, while the genes outside the circle are transcribed in a counterclockwise direction. Genes are color-coded to represent distinct functions, with the dark gray of the inner circle indicating the GC content in the genome and the light gray indicating the AC content. The template map is based on the structure of *A. erecta*.

**Figure 2 genes-14-01894-f002:**
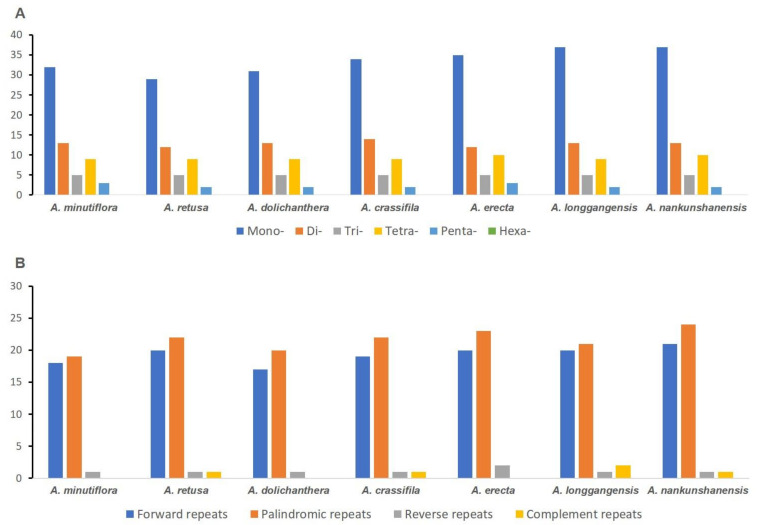
Sequence repeats in chloroplast genomes of the seven *Aspidistra* species. (**A**) Types and numbers of SSRs in the seven plastid genomes. (**B**) Comparison of interspersed repeats in the seven plastid genomes.

**Figure 3 genes-14-01894-f003:**
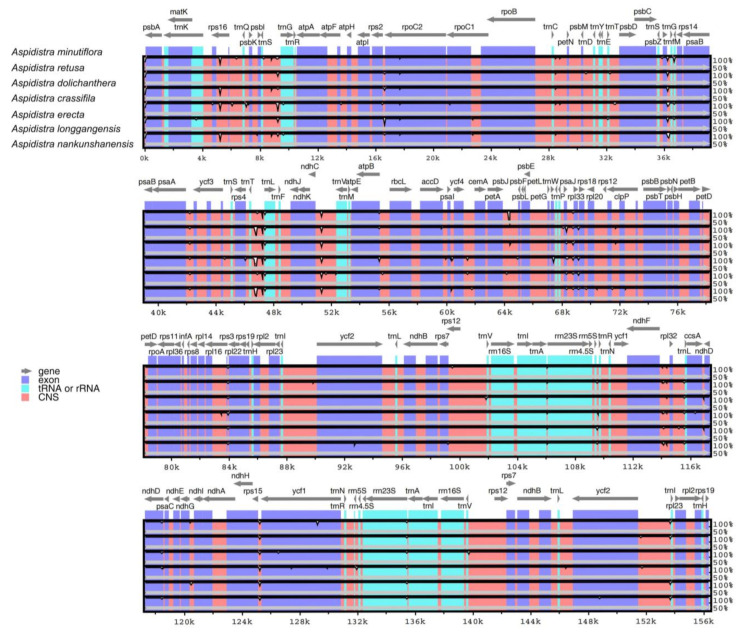
Comparative analysis with mVISTA of the plastomes of seven *Aspidistra* species. The chloroplast genome of *A. minutiflora* was used as a reference. The annotated genes and their direction are indicated by gray arrows. The different regions on the genome are marked with different colors. The vertical scale represents the similarity among sequences.

**Figure 4 genes-14-01894-f004:**
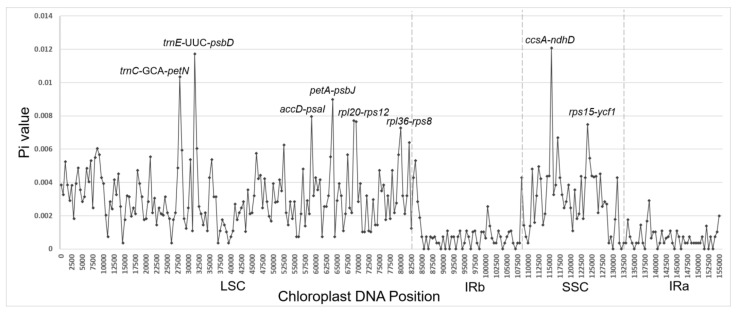
Nucleotide diversity (Pi) of the 11 *Aspidistra* plastomes. Each black diamond represents the Pi value per 500 bp window. A total of eight positions had Pi values greater than 0.007, and all were intergenic regions (*trnC*-GCA-*petN* (0.01033), *trnE*-UUC-*psbD* (0.01171), *accD-psaI* (0.00796), *petA-psbJ* (0.00898), *rpl20-rps12* (0.00771 and 0.00764 for the two adjacent sliding windows, respectively), *rpl36-rps8* (0.00727), *ccsA-ndhD* (0.01207) and *rps15-ycf1* (0.00749)).

**Figure 5 genes-14-01894-f005:**
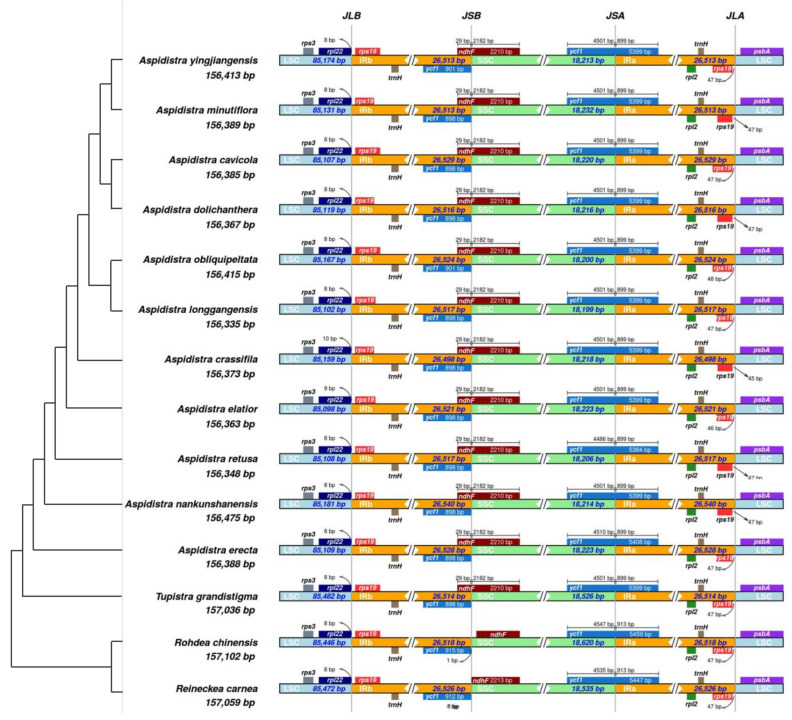
Comparison of the borders of the LSC, SSC, and IR regions of 11 *Aspidistra* species and three outgroups. The tree on the left is simplified from our phylogenetic results (see Section 2.6, Phylogenetic relationships in the Results). The genes near the boundary are shown above and below the mainlines. JLB, JSB, JSA and JLA represent the border loci of LSC/IRb, IRb/SSC, SSC/IRa and IRa/LSC, respectively.

**Figure 6 genes-14-01894-f006:**
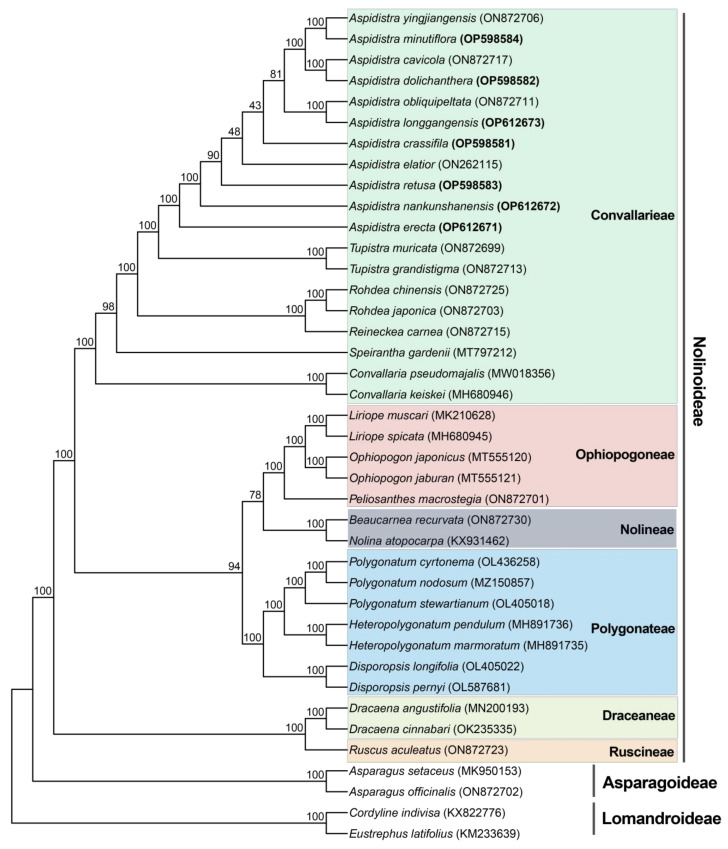
The phylogenetic tree of 40 Asparagaceae species. Species names are shown as tip labels on the phylogenetic tree, with NCBI accessions in brackets. Newly acquired *Aspidistra* plastomes in this study are in bold. Species in different tribes are highlighted with distinct colored backgrounds, with the name of each tribe to the right. Subfamilies are indicated by gray bars. Numbers at nodes are BS supports.

**Table 1 genes-14-01894-t001:** Basic features of the seven *Aspidistra* plastid genomes.

Species	Accession Number in NCBI	Length (bp)				GC Content (%)				Gene Numbers			
		Total	LSC	SSC	IR	Total	LSC	SSC	IR	Total	Protein-coding Gene	tRNA Gene	rRNA Gene
*A. minutiflora*	OP598584	156,389	85,131	18,232	26,513	37.62	35.59	31.61	42.95	130	86	38	8
*A. retusa*	OP598583	156,348	85,108	18,206	26,517	37.61	35.54	31.69	42.97	130	86	38	8
*A. dolichanthera*	OP598582	156,367	85,119	18,216	26,516	37.64	35.58	31.67	42.98	130	86	38	8
*A. crassifila*	OP598581	156,373	85,159	18,218	26,498	37.62	35.57	31.61	42.97	130	86	38	8
*A. erecta*	OP612671	156,388	85,109	18,223	26,528	37.65	35.59	31.66	43	130	86	38	8
*A. longgangensis*	OP612673	156,335	85,102	18,199	26,517	37.64	35.58	31.72	42.98	130	86	38	8
*A. nankunshanensis*	OP612672	156,475	85,181	18,214	26,540	37.65	35.58	31.74	42.99	130	86	38	8

## Data Availability

The data generated in this study are available at Genbank (accession numbers: OP598581-OP598584 and OP612671-OP612673).

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
