# Peer review of "Comprehensive Comparative Analyses of Aspidistra Chloroplast Genomes: Insights into Interspecific Plastid Diversity and Phylogeny"

_genes, 2023, doi:10.3390/genes14101894_

Round 1

Reviewer 1 Report

Huang and co-authors reported study which is entitled " Comprehensive Comparative Analyses of Aspidistra Chloroplast Genomes: Insight into Interspecific Plastid Diversity and Phylogeny”. 

This Paper described 7 Aspidistra chloroplast genomes that have been assembled and analysed. Each Aspidistra plastome has a typical quartile structure similar to that in most angiosperms, highly conserved in GC content, gene number, order and structural characteristics. A comprehensive analysis of these plastomes shows limited interspecific diversity within the genus. In addition, they identified eight relatively hypervariable regions of the Aspidistra chloroplast genome as candidate markers. 

The research was well planned and carried out and it is scientifically sounds. 

Reviewer 2 Report

This study examines the potential of chloroplast genome data for distinguishing species in the genus Aspidistra and inferring their evolutionary relationships. The introduction provides useful evolutionary context to the study drawing on evidence from the literature. The introduction identifies that there is limited data on chloroplast DNA variation to efficiently distinguish species and sets clear aims to expand the dataset with whole chloroplast genomes from seven additional species. The results are summarised clearly and concisely and the detailed figures are intuitive and illuminating. It would be interesting to learn here a little more about the choice of species included in the study. Were they especially threatened or important for other reasons? The discussion places the results within their wider context based on relevant literature in the field. The discussion also points out where the results of this study can help future research such as SSR-based surveys of intraspecific diversity of these species. There is some specific terminology in the discussion about phylogenetic relationships but this will be clear to experts in the field. The methods are also concise and clear. Overall, i find this to be a well-written and focussed paper that needs only minor revision.

Specific comments
L9 Maybe it is worth elaborating a little about karst habitat and why it is special.
L53-57 Break up this long statement for clarity.
L128 Do you mean "numbers of repeats" here?
L160 Should this be "pi > 0.007" based on the values that are subsequently presented.
L186-188 The tree to the left of Figure 5 needs to be explained in the legend.
L191-192 Figure 6 shows more outgroup species in addition to Polygonateae.
L194 Explain the BS acronym at first use. Later, i see that there is a paragraph of used acronyms. Confirm if this is approved journal format.
L267 Drop unecessary "permanently"
L288 The supplemetary information link needs to be corrected here, although it was available for me to inspect as reviewer.
L298-299 Mention the type of manual corrections that were needed.
L300-301 Table S1 does not yet contain the GenBank codes.
L329 Add boostrapping step to the phylogenetic tree methods.

Reviewer 3 Report

The article is interesting because the authors analyzed as many as seven Aspidistra species, including several rare species in terms of plastid genome. However, I believe that the discussion is insufficient. I miss the comparison of the plastid genomes of Aspidistra with representatives of other genera belonging to the Asparagaceae. And yet such data is available, consider the following papers and species: Convallaria keiskei and Liriope spicata (https://www.nature.com/articles/s41598-019-41377-w), https://pubmed.ncbi.nlm.nih.gov/22539521/, https://www.tandfonline.com/doi/abs/10.3109/19401736.2014.953132?journalCode=imdn21, https://esciencepress.net/journals/index.php/JPBG/article/view/2355
